# Plant Cell Walls: Impact on Nutrient Bioaccessibility and Digestibility

**DOI:** 10.3390/foods9020201

**Published:** 2020-02-16

**Authors:** Claire Holland, Peter Ryden, Cathrina H. Edwards, Myriam M.-L. Grundy

**Affiliations:** 1School of Agriculture, Policy and Development, Sustainable Agriculture and Food Systems Division, University of Reading, Earley Gate, Reading RG6 6AR, UK; claireholland1988@gmail.com; 2Quadram Institute Bioscience, Norwich Research Park, Norwich NR4 7UA, UK; Peter.Ryden@quadram.ac.uk (P.R.); Cathrina.Edwards@quadram.ac.uk (C.H.E.)

**Keywords:** cell wall, nutrients bioaccessibility, nutrients digestion, polysaccharides, plant-based food matrix

## Abstract

Cell walls are important structural components of plants, affecting both the bioaccessibility and subsequent digestibility of the nutrients that plant-based foods contain. These supramolecular structures are composed of complex heterogeneous networks primarily consisting of cellulose, and hemicellulosic and pectic polysaccharides. The composition and organization of these different polysaccharides vary depending on the type of plant tissue, imparting them with specific physicochemical properties. These properties dictate how the cell walls behave in the human gastrointestinal tract, and how amenable they are to digestion, thereby modulating nutrient release from the plant tissue. This short narrative review presents an overview of our current knowledge on cell walls and how they impact nutrient bioaccessibility and digestibility. Some of the most relevant methods currently used to characterize the food matrix and the cell walls are also described.

## 1. Introduction

As a staple of the average human diet, plants are a food source rich in a wide variety of nutrients, vitamins and minerals. Plant-based foods, commonly included in the human diet, are vegetables, fruits, cereals, legumes, seeds and nuts. To fully support their growth and health, human beings consume numerous plant parts (Figure 1), including the stems (asparagus), leaves (spinach), inflorescence (broccoli), petiole (celery branches), axillary bud (Brussel sprouts), tubers (potato), roots (carrot), fruit (ripened ovary: tomato, apple), and seeds (almond, wheat and chickpea) [1]. Plants provide humans with essential macronutrients (carbohydrates, proteins and lipids), micronutrients (vitamins and minerals), dietary fibers, and other phytochemicals (e.g., phenolic compounds, and phytosterols) [2]. The nutrient composition and physicochemical properties vary between plant organs, partially due to their differing roles in both the developing and grown plant. For example, roots (e.g., carrot) store energy, often in the form of carbohydrates (starch), while ensuring water and mineral uptake from the soil [3]. Seeds (e.g., almond and wheat) are also packed with macronutrients (starch, lipids and proteins) to sustain shoot development until it becomes a self-sufficient organism. The plant organs are composed of different tissues, such as parenchyma and xylem, and these tissues are made of more than one type of cells. Each cell within the plant is delimited by a cell wall that cements it to its neighbors. Within the living plant, these cell walls have specific chemical, physical and mechanical properties to fulfil a determined function (provide strength and structure to the cell, withstand turgor pressure, and permit interactions with the environment). For instance, parenchyma cells are spherical with thin cell walls while xylem cells are long with thick lignified walls. Once harvested and ready to be consumed, this diversity persists, which has a major bearing on the bioaccessibility and digestibility of nutrients and other phytochemicals contained within the cells. Bioaccessibility is defined as the proportion of a nutrient or food compounds that is ‘released’ from the complex food matrix and potentially available for digestion in the gastrointestinal tract (GIT) [4]. Compound other than cell wall polysaccharides, such as antinutrients, also influence nutrient bioaccessibility and digestion, however these will not be covered in this review.

The plant-based foods we consume are generally classified as cereals, vegetables, fruits, seeds and nuts, and legumes, but this classification does not correspond to any specific botanical grouping. These foods exhibit a wide variety of plant structures and nutritional composition. As mentioned above, they play an important role in human health as they are a source of essential compounds: macronutrients that provide energy, vitamins and minerals, that act notably as cofactors for many enzymes and antioxidants, dietary fibers, and phenolic compounds [5,6,7]. However, plants have developed strategies (synthesis of secondary metabolites and antinutrients) over the course of evolution to protect themselves from predators, such as birds, mammals, or harsh environmental conditions. With the emergence of agriculture, plants species have been cultivated and bred by humans. Certain desirable plant characteristics have been selected so that they become more edible and, consequently, through this process, some of the protective attributes of the plants may have disappeared [8]. Nevertheless, in order to be digested in the human GIT, most plant-based foods need to be processed (e.g., milled, cooked, and/or fermented) [9]. Indeed, the structure, in particular the cell walls (i.e., dietary fibers), of plant-based food matrices can be difficult to break down due to their resistance to the native enzymes of the human GIT.

In many plant-based foods, digestibility (and palatability) depends on reaching a particular stage of maturity [10]. For example, as fruits ripen their cell walls soften, which eases mastication and the release of nutrients from the food matrix; at the same time, the polysaccharides within the fruit are hydrolyzed, leading to the release of monosaccharides, conferring sweetness [9,11]. On the other hand, vegetable shoots and leaves are often harvested at a stage before secondary wall formation since this toughening of the cell walls, caused by acetylation and/or lignification, will decrease the palatability of the food, and hinder the bioaccessibility and digestibility of the cell content [10]. 

In the present work, we discuss how the plant cell wall—an essential plant component—modulates nutrient bioaccessibility and digestion, and describe the main mechanisms by which it plays this role. We also report some of the current methods and technology used to characterize (in term of structure and physicochemical properties) the plant food matrix and the cell walls.

## 2. Cell Wall: The Main Contributor to the Release of Food Constituents

As the key structural component in plants, the cell wall and its diverse properties determine many of the physical and chemical characteristics of the cell; controlling size, strength, rigidity and flexibility. Plant cells, contrary to animal cells, contain a rigid wall that surrounds the plasma membrane, encapsulating macronutrients within the cell (Figure 2). 

### 2.1. Molecular Composition of the Cell Wall

Numerous excellent reviews and detailed papers have been published over the years on cell walls and the polysaccharides that compose them. We refer the reader to some of them [10,12,13]. In brief, the cell wall encompasses an organized network of cellulose microfibrils integrated within a hydrated gel-type matrix typically comprising pectin (homogalacturonan (HG), rhamnogalacturonan I and II (RG-I and RG-II), and xylogalacturonan), hemicelluloses (xyloglucan, mixed-linkage glucan (MLG), mannans, and xylans), and small amounts of glycoproteins, phenolic acids (in particular ferulic acid), minerals and, in some specialized cell-types, lignin [10,12]. This structure has sufficient strength to resist turgor pressure whilst remaining dynamic. 

### 2.2. Organization and Interaction Between Cell Wall Components

While the cell wall composition of most plant-based foods is now known, unraveling the organization of the polysaccharides and other components within the cell wall still remain a challenge [14]. However, evidence reveals that much of the coherence of the cell wall matrix depends upon both the covalent and non-covalent interactions between the polymers within the cell wall [12]. Cellulose self-associates via hydrogen bond formation between adjacent glucose residues on neighboring cellulose chains, resulting in the formation of highly rigid microfibrils (Figure 3). The scaffold formed by the association of these microfibrils provides resistance to the internal osmotic pressure of the cell as a result of its high tensile strength [15,16]. Pectic polysaccharides self-associate through divalent cations and hydrophobic bonding [17], resulting in a gel-like matrix that can impact the porosity of the cell wall and digestibility of macronutrients. Hemicelluloses and pectic polysaccharides can associate with cellulose. Phenolic acids and lignin form covalent cross-links to polysaccharides.

Although cellulose remains ubiquitous as the core structural scaffold of the cell wall, the composition of the matrix phase varies both, chemically and physically from the phylogenetic to the cellular level allowing its adaptation to specific functions and precise regulation [18,19]. The edible tissues of dicotyledonous fruit and vegetables and non-gramineous monocotyledonous plants have cell walls classified as Type I while grass/commelinids have Type II cell walls [20]. The major hemicellulose in Type I walls is xyloglucan [21,22], which comprises around 20% of the dry weight of the plant cell wall whilst pectin contributes approximately 30% (*w*/*w*) [23], introducing strength and flexibility. By contrast, pectin content in Type II walls is less pronounced, whilst xyloglucan content is often very low. For example, xyloglucan content in barley has been recorded as between 2–5% (*w*/*w*) of the total dry weight [24]. In place of xyloglucan, MLG (for example as described in the Poales and Equisetales model systems [25]), glucomannan and glucuronoarabinoxylan (GAX) fulfil similar functions as they are structurally analogous [20,26]. In Type II cell walls, xylan is the main non-cellulosic polysaccharide (predominately arabinoxylan or GAX [24]). Differences in the fundamental structure of the cell walls also suggest differences in specific enzymes, proteins and developmental signals between the two wall types. 

### 2.3. Porosity

The composition and organization of the polysaccharides within the cell wall, in particular the degree of pectin (homogalacturonan) esterification, determine its biomechanical properties and porosity [27]. Cell wall porosity, in turn, influences the diffusion of digestive enzymes and other agents of digestion (bile salts) inside the cell, thereby, affecting macronutrient digestion and the release of their digestion products. Only molecules with a diameter smaller than the pores, the size of which is specific to the plant tissue and organ studied (limiting Stokes’ diameters of about 4 to 5 nm; 4.6 nm corresponding to a 41 kDa dextran), would have the ability to permeate the plasma membrane and reach the cytoplasm [28,29,30]. The size of the pore but also its conformation and flexibility govern the diffusion of a molecule through the cell wall.

Cells, with intact walls, separated from plant tissues are becoming a popular tool to investigate the permeability of the cell walls and macronutrient digestibility [31,32,33,34,35,36,37,38,39,40,41]. These in vitro studies confirmed that digestive enzymes (α-amylase, proteases and pancreatic lipase) cannot diffuse through the cell wall of many plant-based foods (almond, wheat, chickpea, pea, mung bean, red kidney bean, and sorghum) whereas some cell wall appeared to be more permeable (common bean, potato tuber, banana and mango). Fluorescently-labelled probes (i.e., fluorescein isothiocyanate-labelled dextran and albumin) were also utilized to estimate the size of some of the pores of these plants’ cell walls [35,36]. From these studies, it was found that porosity varied greatly between plant species and organs, with almond and legumes having less porous cell walls than potatoes.

The extent to which the cell wall influences the bioaccessibility and digestion, in the upper gut, of intracellular components varies dramatically depending on the physicochemical properties of the plant cell walls and how much of the cellular/tissue/organ structure remains after processing, mastication, and digestion in the upper GIT [42,43]. 

## 3. Bioaccessibility and Digestion of Plant-Based Food Components

Digestion involves the breakdown, through both mechanical and chemical mechanisms of large particles or insoluble food molecules into smaller soluble moieties that can be absorbed by the body. The human digestive system involves a multi-stage processing system in which different nutrients are degraded at different stages by specific digestive enzymes [44] (Figure 4). However for these processes to occur, the enzymes have to be able to come in contact with their substrate which can only be achieved if the latter is bioaccessible—either the enzymes can penetrate the cell wall or the nutrients are released from the food matrix.

### 3.1. Cell Wall Properties and Food Structure

Cell walls are, therefore, the main element that gives the structure of the plant tissue and the foods derived from them [45]. Additionally, by controlling the movement of water in and out of the cells (variation in turgor pressure), cell walls dictate the mechanical characteristics of plant-based foods and thereby their response to mechanical stress (e.g., processing or mastication). The physicochemical properties of the polysaccharides and overall organization of the cell wall determine their behavior in the upper GIT, which in turn affects nutrient bioaccessibility and digestion [43,46,47]. Table 1 shows a summary of the main physicochemical properties specific to plant cell wall polysaccharides. However, it is noteworthy that these properties are interconnected and one effect can be linked to another (e.g., solubility to viscosity or fermentation). 

Plant-based foods that are ingested reach the gastric environment as plant tissue fragments—particles of different sizes and shapes, and compounds released from the food matrix. These fragments contain cells with walls that have various degrees of intactness as a result of processing (e.g., milling and cooking) and mastication. Therefore, despite the intrinsically high nutrient content of numerous plant-based foods, the extent to which these are accessible to digestion within the human GIT varies. Furthermore, depending on whether the polysaccharides are still located within the cell wall matrix or solubilized in the gastrointestinal fluids, their physicochemical properties will differ.

In order to be ediblemost plants, especially cereals and legumes, require some form of processing before being consumed. This includes removal of the shell/ hull, particle size reduction (e.g., milling/grinding), heat treatment (e.g., boiling, drying, baking and roasting) or enzymatic treatment [51]. These processes will disrupt the plant tissue matrix; the cells of a plant can either separate or rupture depending on the cross-links holding the cell wall together and the force applied [9]. The extent of cell wall rupture and/or separation depends on the processing used (type and duration of the treatment) and the fracture properties of the plant tissue [52]. The porosity of the cell wall can also be altered, particularly during cooking [36,37]. Moreover, during processing cell wall polysaccharides are susceptible to modification by the activity of endogenous enzymes, chemical degradation, or by microbial enzymes in fermentation processes. Some of these activities can continue during mastication and digestion [53]. Of the cell wall components, the pectic polysaccharides are most susceptible to modification [54]. For example, pre-cooking can activate pectin methyl esterases and cross-linking of the de-esterified pectic polysaccharides by divalent cations can increase [55,56]. Meanwhile, other processes lead to the breakdown of the pectic network by removal of divalent cations, hydrolysis or β-eliminative degradation, which can result in tissue softening, cell wall swelling, an increase in cell wall porosity and cell separation [9]. 

Despite processing, there are still components of the plant-food matrix that are unaltered during human digestion. Humans have not evolved the necessary enzymes to effectively cleave the glycosidic bond integral to numerous molecules within the plant cell wall; these molecules are commonly referred to as dietary fibers—although the current definition also includes resistant starch and oligosaccharides [57]. Therefore, two fates exist for the cell wall polysaccharides, either they leach out or remain within the cell wall matrix, both having consequences for nutrient bioaccessibility and digestibility [58,59,60]. Because of this, it is necessary to move away from the study of single cell wall polysaccharide, particularly purified forms, and instead focus on more complex food matrices, in order to shed light on the functionality of plant-based foods and their cell wall components [61,62].

### 3.2. Encapsulation

Encapsulation refers to the trapping of nutrients within an intact cell wall, which acts as an impassable or semi-permeable barrier and thereby limits or prevents nutrient bioaccessibility and digestion. Physical alteration of the overall structure of plant-based food, through mechanical processing prior to ingestion, can increase nutrient release. Indeed, particle size reduction of wheat, pea, almond, walnut and peanut, via grinding and milling, was shown to increase the amounts of cell wall ruptured, thereby making the cell content available for digestion [63,64,65,66,67]. 

A recent paper reviewed how various forms of processing impact the release of nutrients and their subsequent digestibility, with carotenoids from fruits and vegetables, and starch from legumes as case studies [68]. The conclusion of their work is that thermal treatment (cooking) is the most effective process to improve carotenoids bioaccessibility as cell walls are then disrupted due to pectin depolymerization and solubilization. An early study, however, revealed that particle size reduction was more important than cooking in controlling carotenoids bioaccessibility [69]. Differences in cooking techniques may explain this discrepancy. Particle size reduction also facilitated starch bioaccessibility and digestibility in legumes. Cooking of legumes promoted pectin solubilization, but contrary to fruits and vegetables, this led to cell wall swelling [34] and/or cell separation [70], similarly to cell wall disassembly observed during fruit ripening [40]. Cell walls that remain intact, such as those of separated cells, will prevent or slow down macronutrients digestion (see Section 2.3). Legumes are therefore particularly resistant to processing, and even after cooking starch digestion has been found to be limited [34]. Indeed, cell walls prevented the gelatinization of starch inside cooked intact, separated cells, thus trapping the granules inside the cells. In addition, to these mechanisms (restricted cell wall permeability and reduction in starch gelatinization), the presence of a protein matrix could also hinder starch digestibility as shown in cereals and legumes since protein and starch compete for water [31,34,71]. The protein is likely to gelatinize sooner and form a continuous gel phase with un-gelatinized starch granules embedded, a structure that may persist even when the cell wall has broken away. 

Nutrient encapsulation can be reduced by dry thermal treatments, such as roasting or baking. Even though the cell walls are likely to rupture during these processes, the nutrients within the cells may still not be fully available for digestion (e.g., coalesced lipids), as observed with nuts [65,72].

Although, some of the undigested cell wall material passes into the large intestine and can be fermented as an energy source for the gut microbiota. The degree of fermentation, however, relies on many factors, including the type of polysaccharides present (cellulose is slowly fermented and lignin is resistant to hydrolysis), the food matrix (accessibility of polysaccharides to bacterial enzymes) and its complexity [73,74]. Fermentation by microorganisms will promote the release of nutrients that escape digestion in the upper GIT [75]. Short-chain fatty acids are produced and absorbed by the host following the degradation of nondigestible carbohydrates (cell wall polysaccharides) by these microorganisms [76]. However, the extent of digestion (by microbial enzymes present in the colon) and absorption of the release nutrients is likely to be limited [39,75].

### 3.3. Solubility and Resulting Properties (Viscosity) 

The solubility of polysaccharides in aqueous environment, such as the gastrointestinal fluids, depends, notably, on their molecular weight, spatial organization, glycosidic linkages, monosaccharide composition and charge [77]. Once in solution, the polysaccharides can swell and/or interact with each other to create networks, which is the case of polymers with random coil conformation, or aggregates [78,79]. By doing so, the components of the plant cell wall can cause changes in the rheological behavior of the GIT content. Indeed, certain soluble cell wall components, such as mixed linkage β-glucan, can alter the local rheology within the GIT, which may increase the viscosity of the stomach and intestine content, thereby reducing mixing of the digesta and resulting in nutrients escaping digestion [48]. However, to have an impact on the viscosity of the gastric or intestinal content, the polysaccharides first have to be released from the food matrix. As an example, despite being “soluble”, only a fraction of the total amount of β-glucan contained within oat matrices was solubilized during simulated digestion [59]. Therefore, when investigating the impact of cell wall components on viscosity, it is critical to consider the matrix into which they are embedded as the mere presence of a cell wall component does not warrant a functional effect [80].

### 3.4. Sequestration 

Sequestration (also referred to as binding) is another mechanism that can lead to decreased bioaccessibility of nutrients. Sequestration can occur at different length scales: interaction with solubilized polysaccharides (e.g., β-glucan and arabinoxylan) and entrapment of digestive agents or substrates into particulates or porous network [81,82,83,84]. At the molecular level, the structural and physicochemical features of the polysaccharides, particularly their charge, hydrophobicity and hydrophilicity, affect their ability to interact with other compounds. For example, cellulose has a hydrophobic binding capacity for certain molecules—e.g., polyphenols [85]. The cholesterol-reducing effect of certain water soluble dietary fibers, such as those found in oat (β-glucan), is proposed to arise from the sequestration of bile salts and excretion of the excess in feces [83,86]. The binding of amylase to cell wall components, and subsequent inhibition of the enzyme, has also been demonstrated in vitro [32,82]. However, in spite of some advances in this area of research, further work is still needed to clarify the interaction(s) taking place in vivo between complex sources of dietary fibres (particles vs. solubilised polysaccharides) and the different agents of digestion (especially digestive enzymes and bile salts).

These studies show that both the composition and organization of the cell wall must be considered in order to understand the behavior of plant-based foods and their cell wall matrices during digestion. Investigating and characterizing (e.g., molecular weight and concentration) one single cell wall constituent (e.g., pectin) is not sufficient to explain the mechanism(s) involved nor to predict the digestibility of plant-based food. It is also clear that there is a lot of diversity among plant tissues and organs, in term of cell wall degradation and overall behavior during processing and digestion in the GIT (swelling, separation and rupture). 

## 4. Characterization of Plant-Based Foods and Cell Wall Materials

There are various techniques and methods that exist to characterize plant-based food materials before and during digestion. These techniques provide some information about the physicochemical properties of the sample, while giving an appraisal of the extent of degradation of the plant tissue and the amount of nutrients or food compounds released. Several length scales (from macro- to micro-/nanoscales) and types of sample preparation have to be considered for the analysis to be meaningful, especially concerning the cell wall [61,87]. A brief summary of the most commonly used techniques is presented below. Additional details about the characterization of cell wall materials, based on research objectives can be found elsewhere [61]. 

### 4.1. Overall Appraisal of the Food Structure

Different techniques can be utilized to provide both an overall assessment of the structure and organization of a plant-based food, and a more informative analysis of the content of specific macro- and micronutrients and other food constituents, both pre- and post-digestion. To fully understand the mechanisms involved during digestion, the characterization ought to be performed on the particles generated as well as the compounds solubilized in the aqueous phase [87]. For example, particle size analysis is often utilized at the tissue- and nutrient-level (e.g., lipid droplets and starch granules) to track the effect of digestion on size distribution and extent of degradation of the plant tissue and its constituents. Rheological measurements are also informative, since the apparent viscosity allows estimation of the amount of dissolved macromolecules/polymers (e.g., soluble polysaccharides and starch). Size-exclusion chromatography (SEC) complements this data by measuring their size and molecular structure [88,89]. Furthermore, viscoelasticity measurements provide relative magnitudes of the “solid-like” and “liquid-like” behavior of the sample. As briefly mentioned above, the rheology of the plant-based food consumed will impact its behavior in the GIT and, as digestion progresses, the viscosity of the food will change.

A range of microscopy techniques are commonly employed to study food structure, including the size and shape of cells within the plant tissue, and the thickness and degree of integrity of the cell walls [70,90,91,92,93]. Structural examination through microscopy also gives information about how nutrients and other food constituents are organized within the food matrix; not just initially to determine the overall structure of the food matrix but also post-digestion to qualitatively evaluate the damage caused, for example to the cell wall, by this process [63,69,94,95]. 

Therefore, different microscopic techniques can be used for different applications, depending on the sample itself (solid, liquid, or living organism), the environment (vacuum, ambient atmosphere, or liquid), the resolution requirements (cellular to nanoscale), and the use of additional probes (e.g., fluorescent labels and antibodies). While light microscopy is typically used to study whole cells or tissue fragments, scanning electron microscopy (SEM) scans the sample surface, providing a topographical image with a resolution of less than 1 nm [96]. Transmission electron microscopy (TEM), on the other hand, allows the internal structure of the sample to be visualized [91]. 

Vibrational spectroscopy, such as confocal Raman microscopy, is a non-invasive technique that also provides information on the chemical composition of the sample including the cell walls [97,98]. Although, Fourier-transform infrared (FTIR) and Raman spectroscopies are powerful tools to investigate the composition of purified cell walls, the analysis of more complex samples can be more challenging [98,99,100,101]. Indeed, despite being able to map different classes of molecules within the sample (when coupled with a microscopic device), the bands corresponding to the carbohydrate region often cluster together making the identification of specific carbohydrate molecules complicated.

More sophisticated techniques allow the study in vivo of digestion and provide some information about the structure of the food within the human GIT. They are, for instance, magnetic resonance imaging (MRI), nasogastric and intestinal intubation to obtain aspirates, and ingestible sensors such as camera capsules [102,103,104,105]. However, some limitations exist with these techniques, including restriction in the number of foods that can be administrated to the volunteers, low resolution (particularly for MRI) and sampling error (appraisal of the entire digesta may not be possible and only particles of a certain size can be recovered by intubation).

### 4.2. Characteristics at the Cellular and Molecular Levels 

To perform the analysis mentioned in this section, cell wall matrices or individual polysaccharides often need to be isolated/extracted beforehand; the quality of the data obtained relied on the extraction method used. The first step consists in quantifying the dietary fiber content of the plant-based food studied. Generally, after drying and removal of the lipids and proteins, the cell wall components are isolated via a combination of enzymatic treatments and precipitation. Thus, a range of methods validated by the Association of Analytical Communities (AOAC) exist to determine total dietary fibers, and/ or differentiate between the water-soluble and insoluble fractions. The AOAC 2017.16 is to date the most comprehensive method that is in agreement with the CODEX definition [106]. In parallel, available carbohydrates, including digestible and resistant starches, could be measured to obtain an overview of the digestible and non-digestible polysaccharides and other carbohydrates contained in the sample [107]. The polysaccharides constituting the cell wall can then be identified with more qualitative approaches, such as gas chromatography-mass spectroscopy (GC-MS), high performance liquid chromatography (HPLC), and comprehensive microarray polymer profiling (CoMPP) [108]. In order to analyze the cell wall components (sugar and uronic acid content), lipids and proteins have to be removed beforehand from the sample. Usually this is performed using Soxhlet extraction and washing with a detergent (sodium dodecyl sulfate, SDS). They give further insight into the specific structure, composition and quality of the individual extracted poly-/oligo-/monosaccharides.

Hydrated cell walls and isolated polysaccharides can be visualized by atomic force microscopy (AFM) [109,110]. AFM produces three-dimensional (3D)-images with a resolution in the order of a fraction of a nanometer. It is not just the structure of the polysaccharides that can be studied, but also their interactions with other components. The glycosidic linkages in a polysaccharide can be determined directly by permethylation, hydrolysis, peracetylation and analysis of the partially methylated alditol acetates by GC-MS [111,112]. The linkages can be deduced from the pattern of acetylation.

Nuclear magnetic resonance (NMR) is another useful technique within the cell wall analysis toolbox, allowing the purity and molecular structure of a sample (whether purified or a mixture) to be determined [14]. It can also be used to identify the physical properties of a sample at the molecular level, including conformational exchange, solubility, phase changes and diffusion, and its physiochemical features, such as cross-linking. NMR studies of never dried walls showed that pectin, rather than xyloglucan, makes up the majority of contacts with cellulose and may be the main mechanical tethers [113].

SANS (neutron) and SAXS (X-ray) are both types of small angle scattering techniques integral to polymer science, allowing probing of macromolecules at length scales from 1 nm to 1 μm to discern the size, shape and spatial relationship of a given sample [114,115]. Information can also be obtained in real time, allowing study of how physical conditions (such as pH and temperature) affect the macromolecules. SANS has been particularly integral to research into cellulose structure and network properties, as well as water interaction with polymers [116].

## 5. Conclusions

Plants are an important source of macro- and micro-nutrients, but the bioaccessibility of these is significantly limited by the cell wall. Dietary fibers, that consist principally of cell walls, and their positive effect on health is now well recognized but the mechanisms by which they exert their action remain largely unclear. Progress has, however, been made in our understanding on how processing modulates the structure of plant-based foods and cell wall permeability. For instance, there are treatments that can be applied to plant tissues and their derived foods, such as cooking and particle size reduction, to enhance nutrients bioaccessibility. Humans consume a wide variety of plant-based foods and have developed overtime a range of techniques to improve their digestibility. Today, we are in a situation where foods have been processed to such an extent that individuals are, in some parts of the world (i.e., Western countries), obtaining too much nutrients and energy from them. With the advances in this field of research, we will, in the future, be able to precisely manipulate the matrix of various plant-based food to achieve a specific effect—enhance or decrease nutrients bio-accessibility to tackle the triple burden of malnutrition.

The methods of analysis described above provide researchers with information about the extent of disruption of the cell wall due to processing and digestion. This can then be linked to data obtained on nutrient release from the food matrix, and thus, identify the mechanisms involved. So far, cell wall components and plant-based foods have mainly been studied in isolation, future research should gradually introduce different ingredients that form a meal to assess how these ingredients interact together and affect nutrients digestibility in more realistic conditions.

## Figures and Tables

**Figure 1 foods-09-00201-f001:**
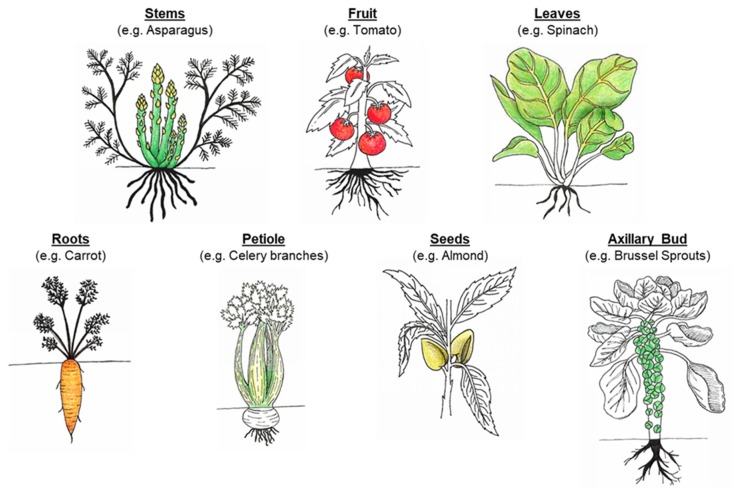
Examples of fruits and vegetables derived from different plant organs/tissues.

**Figure 2 foods-09-00201-f002:**
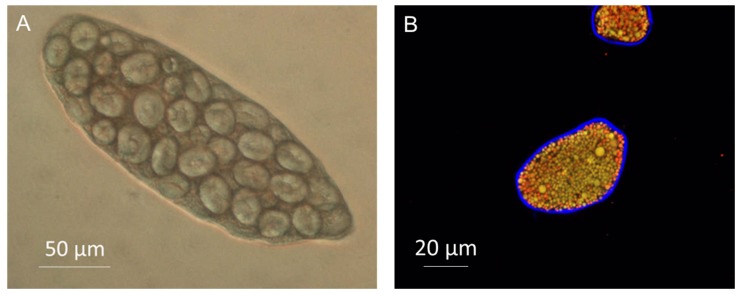
Light microscopy image of chickpea (**A**) and confocal image of almond (**B**) cells. Note the cell wall surrounding the cell full of starch (**A**) or lipid and protein bodies (**B**, cell wall stained in blue with Calcofluor White and lipid in red with Nile Red).

**Figure 3 foods-09-00201-f003:**
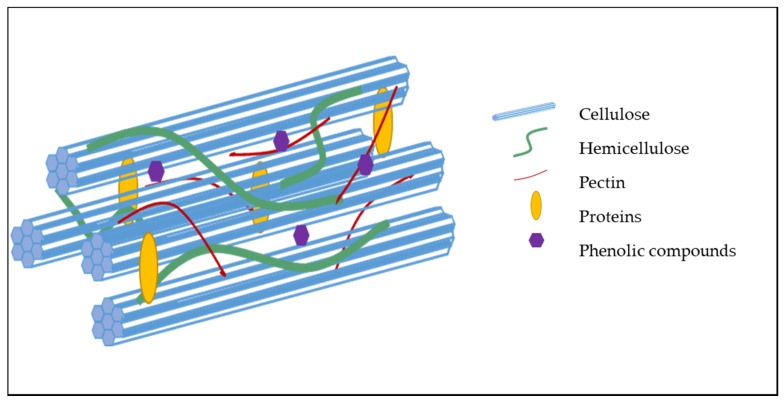
Schematic representation of the structure of a primary cell wall (no lignin) and organization of its constituents.

**Figure 4 foods-09-00201-f004:**
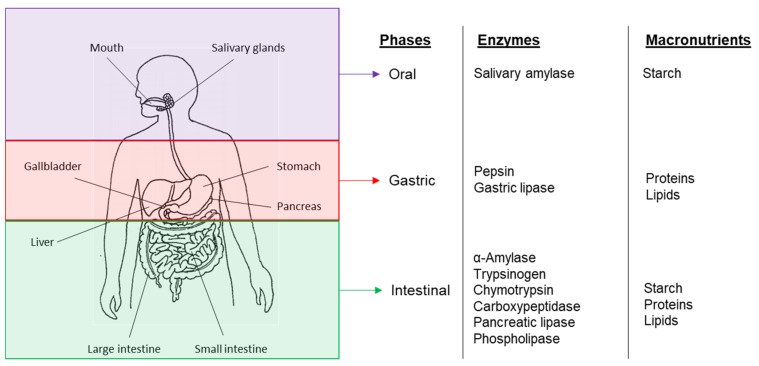
The human digestive system—Diagram of the GIT compartments in which macronutrients are digested and the main enzymes involved.

**Table 1 foods-09-00201-t001:** Main physicochemical properties of plant cell walls and their polysaccharides that may impact nutrients bioaccessibility and digestibility. Adapted from [46,47,48,49,50].

physicochemical Property	Definition	Main Factors Affecting the Property	Dietary Fiber
*Solubility*	Ability of a polysaccharide in a solid form or contained in a solid food matrix to disperse in a liquid (often water) and form a homogeneous dispersion.Polysaccharides (i.e., dietary fibers) are commonly classified as “soluble” or “insoluble” (in the GIT, i.e., an aqueous environment). However, there exists a wide range of solubility beyond these two extremes (poorly soluble, swollen gel-like networks, etc...) [49].	Source (e.g., plant and location within the tissue) Degree of ripening and maturation Processing and food preparation For isolated polysaccharides: extraction and purification method Shape and structure of the polysaccharide (e.g., branching) Polysaccharide molecular weight Medium conditions (e.g., mixing forces, water content, presence of other components, pH and temperature)	*Soluble (in water)* β-glucan Pectins Gums Inulin Some hemicelluloses
			*Insoluble (in water)* Cellulose LigninSome hemicelluloses
*Viscosity*	Internal friction of a liquid, or its tendency to resist flow	Polysaccharide conformation Polysaccharide solubility Particle size Polysaccharide molecular weight Polysaccharide concentration Medium conditions (e.g., mixing forces, water content, presence of other components, pH and temperature) Time	β-glucan Pectins Gums
*Water-holding capacity*	Ability of a fiber source to retain water within its matrix	Particle size Polysaccharide(s) structure Polysaccharide(s) solubility	Cellulose Lignin Some hemicelluloses (e.g., Arabinoxylans)
*Fermentation*	Breakdown of dietary fiber (including resistant starch and oligosaccharides), and some other undigested food components by bacteria in the large intestine.	Polysaccharide solubility Polysaccharide structure and conformation Architecture of the fiber matrix Particle size Microflora	*Rapidly fermented* Pectins β-glucans Arabinoxylans
			*Slowly fermented* Resistant starch Cross-linked pectins Xylans Xyloglucans

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
