# Peer review of "Plant Cell Walls: Impact on Nutrient Bioaccessibility and Digestibility"

_foods, 2020, doi:10.3390/foods9020201_

Round 1

Reviewer 1 Report

This review describes the current knowledge on the cell wall of plants from the human perspective of consuming plant-based foods, which means how this cellular compartment affects the bioaccessibility and digestibility of nutrients in the digestive tract of humans. The introduction points out differences in plant tissues as regards to the structure and composition of the cell wall, which is additionally altered e.g. by the stage of maturity and ripening. The chemical nature of the cell wall is mentioned only briefly with references to previous reviews. Polysaccharides are mentioned as principal cell wall components related to the uptake of nutrients in the digestive tract as their solubility is reflected in the viscosity of stomach and intestine content. The process of food digestion is influenced by the presence of pores in the cell wall, which control the diffusion of digestive enzymes and other agents to their substrates and target molecules, respectively. Resistant polysaccharides are subjected to a microbial degradation in the large intestine. Important chapters are devoted to the processing of plant food such as removal of shells/hulls, milling, and heat treatment (boiling, roasting, or baking). There are several characteristic approaches followed to analyze the structure and properties of a plant food including rheology (to evaluate viscosity) and size-exclusion chromatography (to evaluate size of polymeric molecules). Microscopy techniques are applied to study the size and shape cells or thickness and integrity of cell walls. Magnetic resonance imaging allows to evaluate the content of the digestive tract of volunteers, who consume specific food samples. Analysis on molecular level needs the cell wall components to be isolated and purified first. For example, there is a range of validated methods available to isolate dietary fibers. Structural analyses of molecules utilize e.g. vibrational spectroscopy, NMR or scattering spectroscopic techniques; applicability depends on sample complexity.

There are no objectives from as regards to the content and form. My points to be addressed in a revised version are rather minor:

I miss a figure or scheme accompanying the text part 2.1 and depicting the chemical structure of the building units of relevant polysaccharides such as cellulose, pectin, xyloglucan, lignin etc. to give readers a clear view of structural differences influencing their physicochemical properties.

Line 345, please specify a common procedure for analysis of large polymeric polysaccharides by GC-MS or HPLC. I am not an expert but it seems to me hard if not impossible to get them separated and ionized in this arrangement. Thus I expect a previous hydrolytic treatment for fragmentation is necessary.

Author Response

We thank the reviewer for his/her positive comments on our review. The mchanges made to the manuscript appear in red in the text.

See below our reply to the points raised:

I miss a figure or scheme accompanying the text part 2.1 and depicting the chemical structure of the building units of relevant polysaccharides such as cellulose, pectin, xyloglucan, lignin etc. to give readers a clear view of structural differences influencing their physicochemical properties.

We added a figure (Figure 2) that depicts the structure and organisation of the cell wall and its compounds.

Line 345, please specify a common procedure for analysis of large polymeric polysaccharides by GC-MS or HPLC. I am not an expert but it seems to me hard if not impossible to get them separated and ionized in this arrangement. Thus I expect a previous hydrolytic treatment for fragmentation is necessary.

A sentence was added (lines 351-353) to improve clarity.

Reviewer 2 Report

This study is very interesting, well organized and well written. As the authors mentioned that the plants are important source of nutrients but the bioaccessibility of plants are limited by the strong cell wall. This paper well provided an overview of current knowledge on cell walls and how they impact nutrient bioaccessibility and digestibility. 

 But one thing, it would be better if an explanation is added more on how to improve the digestibility and bioaccessibility rates of plants and how to use them i.e. post-harvest processing, pressure, heating, chemical treatments, changes in plant cultivation methods, etc.

Author Response

We were pleased to receive the comments from the reviewer who find our review interesting and well organised.

But one thing, it would be better if an explanation is added more on how to improve the digestibility and bioaccessibility rates of plants and how to use them i.e. post-harvest processing, pressure, heating, chemical treatments, changes in plant cultivation methods, etc

We agree with the review and this is a very valid comment. Unfortunately, we do not have space here to cover the effect of processing in the level of details that has been suggested. Indeed, each plant and its cell wall will behave differently upon processing methods such as cooking and chemical treatments. For this reason, we kept this concept general as discussed lines 186-201, 215-217 (particle size reduction via grinding or milling), 220-236 (cooking affecting pectin).